# Promoting Well-Being from a Gender Perspective: A Systematic Review of Interventions Using Digital Tools and Serious Games

**DOI:** 10.3390/bs14111052

**Published:** 2024-11-06

**Authors:** Ciro Esposito, Francesco Sulla, Giusi Antonia Toto, Valentina Berardinetti, Andreana Lavanga, Francesco Pio Savino, Salvatore Iuso, Maria Grazia Mada Logrieco, Maria Eugenia Paula Ascorra Costa

**Affiliations:** Department of Humanities, University of Foggia, 71100 Foggia, Italy; francesco.sulla@unifg.it (F.S.); giusi.toto@unifg.it (G.A.T.); valentina.berardinetti@unifg.it (V.B.); andreana.lavanga@unifg.it (A.L.); francesco.savino@unifg.it (F.P.S.); salvatore.iuso@unifg.it (S.I.); maria.ascorracosta@unifg.it (M.E.P.A.C.)

**Keywords:** gender equality, well-being, digital tools, serious games, systematic review

## Abstract

Gender inequalities continue to pose a significant issue across various aspects of life, adversely impacting the well-being of both females and males. These disparities often stem from the ingrained gender stereotypes passed down to young individuals through parental guidance, educational systems, and media portrayal. For this reason, within the psycho-pedagogical field, various intervention models have been developed in recent years, leveraging digital tools to combat stereotypes and enhance well-being among adolescents. The aim of this systematic review is, therefore, to identify studies employing digital tools, particularly serious games, to promote well-being from a gender perspective. The review was conducted using the PRISMA guidelines and collected articles from four databases: Scopus, the Web of Science, PubMed, and PsycInfo. The screening process culminated in the selection of 15 articles. The findings reveal a proliferation of platforms, applications, and programs aimed at promoting well-being by addressing emotional, cognitive (or mental), physical, and sexual health dimensions. Some contributions emphasize nurturing positive attributes within individuals or fostering empowerment as a precursor to well-being. Additionally, certain articles delve into the effect of the COVID-19 pandemic on the well-being of young men and women; in particular, the authors investigated the effect of using an app to improve well-being before and after the pandemic. This systematic review aims to expand the knowledge base on technology-based interventions for social change. It endeavors to empower educators and advance the creation of innovative, evidence-based digital tools that can enhance positive mental health, promote gender equality education, and foster the well-being of young people.

## 1. Introduction

### 1.1. The Roots of Gender Inequality

Gender stereotypes and discrimination against girls and women are still widespread today [1,2,3]. International data reports indicate that gender inequality persists across numerous domains, particularly in the labor sector. For example, the European Union (2020) [4] has highlighted that women face significant inequalities in labor market entry and earnings. Only 67% of women have a job compared to 79% of men, and women’s salaries are about 14% lower than those of men.

The International Labour Organization (ILOSTAT) [5] reports that globally, in 2022, women made up 50% of the working-age population, but only 40% of those were employed, and just 27% held management positions. In 2021, 31% of women and girls aged 15–24 were neither employed, in education, nor in training compared to 16% of young men. In 2019, women were more likely than men to be in informal employment in over 90% of sub-Saharan African countries, 89% of Southern Asian countries, and nearly 75% of Latin American countries. Globally, 22% of working-age women engaged in unpaid care work full time compared to just 1.5% of working-age men. At this rate, the gender gap in unpaid care work is projected to persist until 2228 (or for another 205 years). Additionally, the global gender pay gap stands at an estimated 20%.

Moreover, salary inequality is indicative of some of the main forms of discrimination faced by women. First, women are often “segregated” into lower-paying jobs, such as in care and education [6]. Second, women experience more work–family conflict and are often forced to work shorter hours (and therefore earn less) than men, as they are frequently designated as the primary caregivers and responsible for managing family duties [7]. Third, women face the “glass ceiling” effect [8], meaning they have difficulty in accessing prestigious positions; for example, less than 10% of the CEOs of large companies are women [4].

These inequalities derive from the creation and propagation of gender stereotypes, which assign different social roles and lead to asymmetrical relationships between men and women, typically to the disadvantage of women [9,10]. From this perspective, gender inequality can be traced back to adherence to traditional social rules and gender roles that attribute to men a “manly” type of masculinity, which is expressed through power and control over others [11].

Extreme consequence of this relational asymmetry and this construction of “toxic” masculinity [12] constitute a double risk for women in terms of psychological well-being and physical health. Masculinity has negative consequences on the well-being of both men and women [13], and these further push to perpetrate gender inequalities. Furthermore, the exercise of power associated with the manifestation of masculinity can lead to the implementation of violent discriminatory behaviors against women [14,15].

Given the risk of such a dangerous drift, many scholars have investigated gender dynamics and proposed interventions and strategies to prevent violence and promote gender equality [16,17]. However, interventions aimed at promoting gender equality often focus solely on understanding and changing individual characteristics while neglecting the comprehensive examination of contextual factors. These factors can profoundly influence social relationships and the quality of life for both men and women [18].

Among the still too few contextual factors considered, a major role is played by education: violence against women and prior to that, gender stereotypes and discrimination, which underpin social inequality, have deep cultural roots and are often transmitted through the educational models proposed to young people by parents [19,20], teachers [21,22], and the media [23]. These sources are key influences in the process through which children associate certain roles, behaviors, and even objects with different genders, known as gender typing [24]. These beliefs and attitudes begin to develop in childhood through gender typing and become more entrenched during adolescence [25].

Being implicitly or explicitly taught that construction toys are primarily for boys, for example, can lead boys to develop their spatial visualization skills more than girls, with can impact their academic abilities. Male superiority in mathematics is generally limited to performance in geometry, a branch of mathematics that requires spatial visualization skills [26]. In fact, girls tend to outperform boys in computational skills, and there are no gender differences in performances on tests of basic mathematical knowledge or algebra [27]. Moreover, boys’ superiority in mathematics typically emerges during their high school years, likely due to the lower expectations from teachers and parents regarding girls’ mathematics skills [26]. This can significantly influence girls’ self-efficacy. Indeed, girls still choose STEM (i.e., Science, Technology, Engineering, and Mathematics) educational pathways to a lesser extent than boys [28].

The under-representation of women in these fields may have significant consequences in terms of gender inequality and contribute to the gender gap in the labor market, as previously mentioned.

### 1.2. Well-Being and Gender Inequality

Before delving into the consequences of gender disparities on the well-being of men and women, it is necessary to point out that the term well-being is not used univocally in the literature. Over the years, different authors and approaches have provided different definitions.

Starting from the hedonic perspective, Diener et al. [29,30] discuss subjective well-being, defining it as an individual’s overall evaluation of their life based on four components (two emotional and two cognitive): negative emotions and positive emotions; life satisfaction, and, finally, satisfaction in specific domains, such as work, family, and friends. From the eudaimonic perspective, Ryff [31,32] defines psychological well-being as optimal psychological functioning based on the satisfaction of six dimensions: self-acceptance, positive relationships with others, autonomy, control over one’s environment, a sense of purpose in life, and continuous personal growth. In the field of positive psychology [33], we then find the PERMA model proposed by Seligman [34], which defines well-being as a state of optimal functioning based on five factors: positive emotions, engagement, relationships, meaning, and compliance.

A common criticism of previous conceptions of well-being is that they place responsibility solely on the individual for their level of well-being, overlooking the contextual and social conditions that either facilitate or hinder its development [35,36,37]. To address this issue, [38] proposed the construct of social well-being, which assesses public and interpersonal criteria for good psychological functioning. This construct comprises five factors: Social Integration, understood as the feeling of being part of and receiving support from the community; Social Acceptance, referring to trust in others and maintaining positive attitudes towards others; Social Contribution, defined as the feeling one has of giving something valuable to society; Social Actualization, referring to confidence in the future and positive development of society; and Social Coherence, which is the perception that the world is comprehensible, predictable, and controllable.

Finally, the community psychology approach should be considered [39,40,41]. This approach aims to reduce the social discomfort of individuals and groups and promote well-being through specific methodologies that encourage individuals to reflect on their own life context and implement active behaviors to change the uncomfortable situation they experienced. In this approach, the fundamental concept for the promotion of well-being is in fact that of empowerment [42,43] or that of the process of acquiring power through which the individual’s capacity is increased to actively control their life.

In this approach, the major contribution to the definition of well-being comes from Prilleltensky [44], who defines it as “a positive state, brought about by the simultaneous and balanced satisfaction of the different objective and subjective needs of individuals, of relationships between people, organizations and communities” (p.2). Prilleltensky [44] proposes an ecological model, with well-being being composed of four dimensions: personal, interpersonal, organizational, and social.

Starting from this ecological perspective [45,46,47], we can see that such inequalities can have repercussions on well-being, with different consequences for men and women. Considering well-being in an ecological and multidimensional sense [47,48,49,50] involves evaluating the individual and contextual factors that act in different domains of people’s lives, such as mental health issues, interpersonal relationships, and the occupational field, as well as the social context.

In terms of individual mental health, the literature reports some gender differences. Women tend to report internalized disorders, such as anxiety, depression, and various forms of psychological distress, more often than men [51,52,53]. In contrast, men exhibit higher rates of substance abuse and antisocial disorders (externalizing disorders) [54].

From a relational and social point of view, the literature reports controversial data on interpersonal well-being. On the one hand, men seem to have a better quality of life thanks to strong relationships in formal groups [55]. On the other hand, it seems that women tend to take more care of social relationships within the local community, thus obtaining greater social support and well-being [56].

Finally, regarding the occupational domain, studies [57,58,59] indicate that employment status is strongly correlated with well-being. This means that due to stereotypes, women often face fewer career opportunities and, consequently, experience lower levels of well-being compared to men.

### 1.3. Educate Young People Through “Their” Tools: The Use of Digital Tools and Serious Games

As mentioned, the formation of gender stereotypes begins in childhood and solidifies during adolescence. At this stage, young men and women, as they refine their gender identity and their image of masculine and feminine, may internalize gender inequalities as common and normal, continuing to perpetrate sexist stereotypes, attitudes, and behaviors towards women [60].

The rise of technology presents a unique opportunity to address social inequalities and improve mental well-being among young people, particularly leveraging the tools they engage with daily. Adolescents extensively use digital tools, such as social networks, smartphone apps, and video games [61], across various aspects of their lives, including education [62,63]. A key feature of these digital tools is the possibility to experiment with new situations and develop new skills in an environment that involves a low level of risk in the case of failure, a benefit that is especially evident with serious games.

Serious games are tools designed for players but serve serious objectives, such as educational purposes or skill acquisition [64]. They offer the opportunity to practice, make mistakes, and try again without real world consequences, allowing users to operate in a safe environment [65]. Another important aspect of serious games lies in their ability to immerse users in fictional yet plausible scenarios [66]. When playing a serious game, users may enter a state of complete absorption, commonly known as the “state of flow” [67], which is a critical factor in achieving successful learning outcomes through gameplay.

Building on this premise, the development of serious games has been promoted in the field of educational sciences [68,69]. These playful tools aim to educate players through playful engagement. Specifically, the process of gamification has proven effective in increasing users’ motivation to learn [70] and enhancing learning rates and skills development. Additionally, the use of serious games that incorporate virtual reality paradigm has been associated with higher user satisfaction in these experiences [71].

The use of serious games with young people can increase their awareness of some sensitive issues: well-known is the positive effect in the affective sphere [72,73]. The player needs to deepen their experiential learning, often using role-playing techniques that enhance self-awareness and awareness of others. This encourages them to consider multiple perspectives, giving the possibility to overcome personal interpretation and take into account the feelings of others [74].

Given these premises, it seems reasonable to question whether serious games can also be a useful tool for promoting young people’s well-being from a gender perspective. Indeed, they have already been used to improve gender equality education [75] and promote healthy gender relations, fostering the empowerment of men’s psycho-social well-being and (especially) women’s [76].

### 1.4. Aim

Based on the literature, we conducted a systematic review aimed at identifying digital tools, particularly serious games, that had been developed and used to promote the well-being of adolescents and young adults from a gender perspective. Specifically, the aim was to examine studies that have dealt with well-being understood in its various facets that are present in the literature: the emotional and cognitive components; well-being understood as mental health; well-being related to empowerment, physical, and sexual well-being; well-being from the perspective of positive psychology; and well-being related to a critical event, such as the COVID-19 pandemic.

This is crucial for addressing significant developmental issues such as the gender gap, inequality, and identity. By focusing on these aspects, we can better understand how digital tools can be leveraged to address and mitigate gender-specific challenges, thereby contributing to more equitable and effective interventions for promoting well-being.

## 2. Methods

This systematic review was conducted following the Preferred Reporting Items for Systematic Reviews (PRISMA) [77,78], including 15 research articles published between 2018 to 2022.

### 2.1. Eligibility Criteria

These results were then screened according to the following inclusion and exclusion criteria: The inclusion criteria were as follows: research papers; articles published in the last 5 years (from 2018 to 2022); and written in English. On the other hand, the exclusion criteria were the following: conference papers, abstracts, and letters to editors and studies carried out in fields of study distant from Psychology and Pedagogy, such as Management, Sport Sciences, Economics, and Virology.

On the other hand, no restrictions were set on the type of design used by the research articles nor on the geographical area in which the studies were carried out.

### 2.2. Information Sources

Starting from these criteria, we formulated the following search string: ((school OR scholastic OR adolescent* OR teen* OR young* OR student*) AND (“digital tool*” OR “serious game*” OR game* OR ITC OR APP* OR application* OR digital*) AND (girl* OR woman OR women OR female* OR gender* OR sex*) AND (wellbeing OR well-being OR wellness OR thriv* OR flourish* OR empower* OR welfare)).

This string was used in four electronic databases: Scopus, the Web of Science, PubMed, and PsycInfo.

The first search was carried out at the end of February 2023 and resulted in 3296 results for Scopus, 1212 for the Web of Science, 269 for PubMed, and 27 for PsycInfo.

### 2.3. Search Strategy, Selection Process, and Data Collection Process

First of all, based on our objective and what emerged from the literature explored, we defined the search terms to be used to search for articles related to the topic. These terms were defined according to PICO (Population, Intervention, Comparison, and Outcome) criteria. As the population, we set “adolescents and young people”; as an intervention, we set the use of a serious game or digital tool in general; for the comparison, we used words that refer to gender differences; and finally, as an outcome, we set well-being and empowerment.

At the end of this selection, the identified articles were 1239: 781 from Scopus, 162 from the Web of Science, 269 from PubMed, and 27 from PsycInfo. After the detection of duplicates (*n* = 259), we proceeded to analyze 980 records based on the title, keywords, and abstract. These records were analyzed by a team of four independent researchers with a broad background on the topics explored, who selected them for relevance and eligibility based on their relevance to the objective of the systematic review.

At the end of this selection process, 941 records were excluded due to being categorized as “off topic” with respect to the research topics and objective or with a “wrong population”. The articles defined as “off topic” were those that presented research that was not in line with the objective of this systematic review. For example, we excluded articles that did not actually refer to the development or use of digital tools for the promotion of well-being or because they generally evaluated the effect of the excessive use of social networks or video games on health. Regarding the “wrong population” criterion, articles reporting research carried out only with children, adults, or elderly people were excluded.

At this point, the research team proceeded to read the full texts of the 39 remaining articles, which were examined based on the content and quality of the proposed research.

Regarding the content, 24 articles were excluded due to the following reasons: 9 = wrong population (i.e., no adolescents); 13 = wrong outcome (i.e., no digital tools used to develop some aspects of psychological well-being; and 2 = wrong topic.

### 2.4. Study Risk of Bias Assessment

The quality assessment of the remaining records (*n* = 15) was performed using the Critical Appraisal Skills Program checklists (CASP, 2018) for a prevalence study, cohort study, randomized controlled trial, and qualitative study. All records were evaluated by two independent researchers, who verified that they reported a low risk of bias.

Furthermore, to further verify the degree of agreement between the evaluators, the Cohen’s index k was calculated, which resulted as equal to 1, indicating a perfect agreement between the evaluators and a minimum risk of bias [79].

Therefore, at the end of the entire screening process, 15 records were included in this systematic review. The entire process of article selection is shown in Figure 1.

## 3. Results

As shown in Table 1, the studies included in the review varied widely in the research design used, the geographic area in which they were conducted, the participants recruited, and the well-being-related variables they considered.

Regarding the design, four studies [80,82,85,94] employed randomized or a quasi-randomized controlled trials (RCTs), four [81,87,90,92] used a longitudinal cohort or pre–post design, three [83,86,91] utilized qualitative methods, three were mixed-methods studies [88,89,93], and, finally, one [84] used a cross-sectional design.

In terms of geographical area, the studies were carried out in diverse and distant countries, distributed over almost all continents. Six studies [82,84,86,87,90,93] were conducted in Europe, two [80,85] in the United States, two [81,89] in Central or South America, two [83,91] in Australia (Oceania), one [88] in East Africa, and, finally, one [92] involved participants from various regions of the world.

Regarding the participants, the studies present some differences in relation to the two most relevant demographic variables for the purposes of the study: age and gender.

For age, six studies [80,82,87,88,89,93] involved adolescents aged 12 to 19, five [81,84,85,90,92] included young adults aged 20 to 35 years, and four [83,86,91,94] recruited participants from both age groups.

Regarding gender, two studies [81,86] involved entirely female samples. In eight studies [80,84,85,87,90,92,93,94] the majority (>60%) of the participants were female. In three studies [82,88,93] the percentage of males and females was almost the same (from 47% to 53% of females). Only one study [83] had a noticeably lower percentage of females (33%) compared to males. Lastly, the study by Morgan et al. [91] included participants who were transgender and gender diverse (TGD) young people. Aside from the latter, the other included studies did not delve into non-dichotomous gender categories and did not detail whether they included the category “other” as a possible answer to the question “gender you identify with.”

Finally, among the selected studies, there was considerable variability regarding the dimensions of well-being that the digital tools aimed to improve. Some studies [80,82,91] focused mainly on cognitive components (such as self-efficacy and adaptive strategies) or affective (such as emotions) components of psychological well-being. Other studies [83,89,94] examined aspects of mental health, such as depression or anxiety. Three studies [84,86,93] focused on the development of empowerment, while two of them [81,88] considered factors related to physical or sexual health. Two studies [85,92] explored subjective well-being within the theoretical framework of positive psychology. Finally, two studies [87,90] investigated the effect of the recent COVID-19 pandemic on the psychological well-being of young people.

The results found in the articles will be discussed in the subsequent paragraphs based on the main variables considered.

### 3.1. Cognitive and Affective Components of Well-Being

Starting from a multidimensional view of psychological well-being [95], firstly, we categorized the results based on whether they focused on the general evaluation of life satisfaction, therefore considering the cognitive component of well-being, or whether they examined well-being as a state positive emotional, i.e., considering the affective component.

Considering the cognitive component, Bickman and colleagues [80] developed Screenshots, a school-based program designed to cultivate positive digital social skills in middle school students with the long-term goal of improving their health and well-being. The authors hypothesize that this goal will be achieved by increasing students’ knowledge of online behaviors and changing their beliefs, which, in turn, will positively influence their intended behavior in online social interactions and conflict resolution.

The results showed an increase in knowledge of key curriculum concepts (F = 9.97; *p* = 0.003). The participants decreased (F = 9.00; *p* = 0.004) their beliefs in favor of rude online behavior and increased (F = 4.39; *p* = 0.04) their beliefs in favor of online safety and privacy behavior. The authors also found significant differences with regard to gender. In fact, boys decreased their likelihood of using verbal aggression (F = 5.77; *p* = 0.006) and increased their use of a more prosocial approach to conflict resolution (F = 3.65; *p* = 0.04).

Considering the affective component, Morgan and colleagues [91], in a qualitative study, through the use of focus groups, aimed to understand the role of avatars in games, their impact on the well-being of transgender and gender-diverse (TGD) young people, and their specific needs regarding avatar design. Their results show that for many participants, the creation and/or selection of an avatar that reflects their experienced gender represents a “safe place” in which to begin “testing” and understanding their gender identity before doing so in the real world. The avatar plays an important role not only in understanding but also in “refining” the participants’ gender identity. The participants reported experiencing positive emotions when playing as an avatar created based of their experienced gender, considering it as the only means of their gender expression before coming out and attributing it a potentially therapeutic role during an emotionally difficult time. Positive emotions were also reported when avatars were recognized and accepted in their experienced gender within the game, such as during the interaction with other online players.

The study by Cejudo and colleagues [82] considered both the cognitive and affective components. They conducted a quasi-experimental study evaluating an intervention program using a video game called Aislados (isolated), which was designed to improve the well-being, mental health, and emotional intelligence of adolescents. The results show an increase (F = 1.99, *p* = 0.038, and d = 0.22) in health related to quality of life (i.e., the cognitive component of well-being) and an increase (F = 2.932, *p* = 0.041, and d = 0.38) in positive affectivity (i.e., the affective component of well-being).

### 3.2. Mental Health

With reference to studies that have dealt with well-being intended as mental health, Cheng and colleagues [83] conducted a qualitative study, using participatory research workshops and interviews, to determine the optimal representation of MindMax. This app aims to provide psycho-education for improving men’s mental health, attempting to create a community in order to tackle help-seeking barriers. The app incorporates gamification, mini-games, and social connection. The aim of the investigation was to identify the best way to structure and present MindMax. The workshops and interviews conducted based on user experience indicated a positive reception among users for the integration of mental health, well-being, sports, and video games within the MindMax platform. The participants expressed a preference for alternative modes of information delivery, such as video subtitles and transcripts, while stressing that videos were frequently skipped. This underscores the importance of presenting key information in diverse formats to minimize repetitiveness. The participants also showed sensitivity to the tone adopted by health and well-being apps, with a formal tone perceived as intimidating and an informal tone considered less pressuring but potentially undermining the content’s seriousness. Active engagement and the practice of useful skills were favored. For instance, activities requiring more effort, such as guided meditation and writing a postcard, received better feedback compared to those with minimal effort, like posting a selfie or selecting a quote.

In another study, Wiljer et al. [94], through a RCTs, evaluated the impact of the app Thought Spot, a tool that assesses help-seeking intentions for mental health and the well-being of post-secondary students (17–29 aged). The mixed-effects model found a significant time effect (F = 23.52; *p* < 0.001; f = 0.21) but no significant interaction between the group and time (F = 0.85; *p* = 0.43; f = 0.04). The effect of gender was significant with regard to help-seeking behavior from formal and/or informal sources. In fact, women showed higher scores with regard to formal help seeking (OR = 1.86; *p* = 0.001) and lower scores with regard to informal help seeking (OR = 0.80; *p* < 0.001), compared to men. Similarly, with regard to attitudes towards healthcare professionals, women show lower scores with regard to the tendency to seek professional psychological help (OR = 0.80; 95%, *p* = 0.003).

Finally, in Colombia and Chile, Martinez et al. [89], developed the Cuida tu Ánimo (CTA—Take care of your mood) program, a digital tool designed to prevent and intervene early in cases of adolescent depression. The authors conducted a pilot study with 517 adolescents that used the program, receiving 16 emails for mood monitoring. The study collected both quantitative and qualitative data. As for quantitative results, the use of the CTA website was moderate, with women being more likely to use it extensively (OR = 2.9). The results showed a reduction in depressive and anxious symptoms in adolescents in the Chilean sample (*p* < 0.001), while there was no significant change in the level of symptoms in adolescents in the Colombian sample (*p* < 0.650), highlighting how context is also an element that should not be underestimated. Satisfaction with the program was generally high (82.5%), with most participants finding it useful, easy to use, and comfortable. As for qualitative results, the participants saw the program as a source of support for emotional well-being and a platform for freely expressing themselves and expressed a desire for more interaction with the team.

### 3.3. Empowerment

Three out of the selected articles reported the results of studies that had the aim to increase well-being through empowerment.

A qualitative study conducted by Denton-Calabrese and colleagues [86], using semistructured interviews, examined changes in the self-concept of participants that utilized “go_girl: code + create”, a multidisciplinary information technology program that was based on the needs of young women aged 16–24 who were not in education, employment, or training (NEET). The program aimed to help marginalized young women clarify and broaden their aspirations and develop their digital skills. The participants reported that the program appears to be positively connected to their self-concept related to both education and career aspirations, clarifying their goals, and expanding their vision of what is possible to achieve in their lives. In addition, some participants reported changes in their self-concept related to information technology, beginning to perceive themselves as creators and not just as users of technology.

Ravaccia and colleagues [93] conducted a mixed-methods study, with feedback questionnaires completed at two time points and semistructured interviews, in order to investigate the experiences of young people in the use of MeeToo (which has changed their name to Tellmi), an anonymous, fully moderated peer support tool focused on mental health for young people (aged 11–25 years). The digital tool was created with the aim of fostering well-being through empowerment. The results show that the overall well-being levels of female participants increased from an average of 1.83 to 2.00, reporting a significant increase following the use of the tool (t = 2.15; *p* = 0.04). The semistructured interviews revealed that anonymity helped create a safe space in which to express themselves freely by allowing the participants to connect with others with similar experiences and to feel empowered in providing support to others.

Finally, D’Aprile et al. [84] conducted a cross-sectional study in which 124 young Italian diabetics (mean age = 14.12 years) were involved in the use of Tako Dojo. Tako Dojo was a prototype health serious game that aimed to help patients manage their disease responsibly and improve their quality of life. The results of the study showed a significant correlation between gender-related behaviors and diabetes therapy adherence (r = 0.19; *p* < 0.05), between game experience and empowerment in diabetes management (r = 0.28; *p* < 0.01), and between empowerment in diabetes management and behaviors related to therapy adherence (r = 0.52; *p* < 0.01). It was found that play experience significantly increased empowerment in diabetes management in the participants (β = 0.88; *p* < 0.01). Thus, patient empowerment positively influenced the behaviors related to therapeutic adherence of diabetics (β = 0.46; *p* < 0.01). Regarding gender, it was found that in males, the serious game contributes to developing higher levels of empowerment (β = 1.88; *p* < 0.01) and therapeutic adherence, both indirectly (β = 0.55; *p* < 0.01) and directly (β = 1.15; *p* < 0.01), compared with females. Female participants, on the other hand, showed a positive relationship between empowerment and behaviors related to therapy adherence (β = 0.46; *p* < 0.01).

### 3.4. Physical and Sexual Health

Among the studies included in the review, two of these focused on the physical and sexual aspects related to well-being.

On one side, Cantisano et al. [81] developed the ePsiconut program, a digital tool aimed at improving physical and mental health. In a pre–post study involving 16 young women aged between 18 and 24 years (M = 20.69; SD = 1.74), the authors found that the application of information technology in education can be an innovation, enabling effective behavioral change and knowledge acquisition. Following the ePsiconut program, the participants reported a notable improvement in the overall dietary quality of the meals (t = 0.92; *p* < 0.05). Generally, there was a notable increase in the consumption of “nutritious foods”, such as vegetables (*p* = 0.005), alongside a decline in the intake of “unhealthy foods”, such as sugary drinks (*p* = 0.042). Moreover, their findings revealed a statistically significant increase in healthy lifestyles (t = 3.02; *p* = 0.009), comparing participants’ scores before and after the participation in the program. Finally, subjective well-being showed a significant increase (*p* < 0.05), while anxiety levels exhibited a significant decrease post-program (*p* < 0.05). Most participants rated the program as excellent, noting a substantial impact on their subjective well-being (M = 4.38; SD = 0.72; range: [1;5]).

On the other side, Haruna et al. [88], focused on adolescents’ sexual health education in lower secondary schools, showing that a long-term unhealthy lifestyle and poor use of emotional regulation techniques could damage mental and physical well-being. The authors conducted a mixed-methods study in one public school in a city in Tanzania to evaluated Game-Based Learning (GBL) and gamification interventions and to compare their effectiveness with the existing traditional teaching methods. They invited 120 lower secondary school students, aged between 11 and 15 years, to participate in focus group interviews to collect feedback on their experiences with the three teaching methods (40 students utilized gamification, 40 used game-based learning, and 40 constituted the control group). The mean post-test scores for Game-Based Learning (GBL) (M = 79.94; SD = 11.169) and gamification (M = 79.23; SD = 9.186) were significantly higher (F = 54.75; *p* = 0.001) compared to the control group (M = 51.93; SD = 18.705). Across the constructs of motivation, attitude, knowledge, and engagement, statistically significant differences (*p* ≤ 0.05) were observed, demonstrating that the instructional approach used in the GBL and gamification experimental conditions had a greater positive impact on students’ sexual health knowledge gains and understanding than students in the traditional conditions.

### 3.5. Positive Psychology

Two of the selected articles addressed well-being using the theoretical framework of positive psychology. Positive psychology aims to study and enhance positive aspects of human experience and well-being, focusing on strengths, virtues, and positive emotions. Its goals include promoting overall well-being, cultivating positive emotions and relationships, and optimizing human potential [96].

The first article [85] presented a quasi-experimental study that demonstrated a significant correlation between increased hope and smartphone app use in a group of American university students. The participants utilized the free smartphone app RealLife EXP. After completing the initial assessments, the students in the intervention group were provided with a 28-day mobile app program, which featured 1 to 3 randomly timed hope notifications delivered from Monday to Saturday, as well as constant access to peer stories of hope. In contrast, the control group did not receive any spontaneous notifications or have access to peer stories of hope within the app. Between days 29 and 31, participants in both groups completed post-test measures. The participants seemed actively involved in the intervention and reported it as user-friendly, beneficial, and enjoyable. In comparison to the control group, those who received the intervention exhibited notably higher levels of hope (F = 4.24; *p* = 0.05) compared to those in the control group. However, there were no differences between the two groups regarding hedonic and eudaimonic well-being. There are also no differences based on gender.

In the second article, Parks and colleagues [92] compared changes in well-being over time after using a digital app in users with and without a chronic condition. The study collected data from Happify, a digital well-being program available via website or mobile app. Happify users engaged with specific tasks addressing issues, such as overcoming negative thoughts, through games and activities grounded in positive psychology, Cognitive Behavioral Therapy, and mindfulness principles. The sample consisted of 821 users exposed to Happify for at least 6 weeks (ranging from 42 to 179 days) who met the inclusion criteria. Among them, 450 reported having a chronic condition (e.g., arthritis, diabetes, insomnia, multiple sclerosis, chronic pain, psoriasis, and eczema), while 371 did not. At baseline, users with a chronic condition had significantly lower well-being (M = 38.34; SD 17.40) than those without a chronic condition (M = 43.65; SD 19.13). However, change trajectories for users with or without a chronic condition did not differ significantly; both groups experienced similar improvements in well-being. The authors identified effects for the time from baseline (b = 0.071; *p* < 0.01) and the number of activities completed (b = 0.03; *p* < 0.01), along with a two-way interaction between the number of activities completed and the time from baseline (b = 0.0002; *p* < 0.01). This interaction indicated that completing more activities over longer periods resulted in improved well-being scores. The study’s data support the conclusion that users with a chronic condition demonstrated a significant improvement over time. Despite starting with lower well-being, their change trajectory while using Happify was comparable to that of users without a chronic condition.

### 3.6. COVID-19

The last two articles included [87,90] were two longitudinal studies, conducted, respectively, before and after the COVID-19 pandemic. Both studies focused on the use of a serious game called Grow It!, which is a multiplayer mHealth tool that combines the Experience Sampling Method (ESM) with gamified Cognitive Behavioral Therapy (CBT).

In the first study [90], interventions were assessed using the Grow It! app to prevent mental health problems in adolescents by improving cognitive–behavioral coping, with challenges inspired by therapy. The study was conducted on two independent cohorts: in cohort 1, users played the Grow It! app for six weeks, and in cohort 2, the users played for three weeks.

The results showed contrasting results regarding the effects of the digital tool on emotional well-being. In fact, the authors found that for some participants, there was an increase in positive affects over time (cohort 1: *n* = 308 (64.7%); cohort 2: *n* = 586 (72.0%); b = 0.06 (*p* < 0.0001) for the first cohort, and b = 0.10 (*p* < 0.001) for the second cohort), while for others, a reduction in positive affects was found (cohort 1: *n* = 168 (35.3%); cohort 2: *n* = 228 (28.0%); b = −0.27 (*p* < 0.0001) for the first cohort, and b = −0.52 (*p* < 0.0001) for the second cohort). The same was found for negative affects: some of the participants showed a decrease (cohort 1: *n* = 393 (82.6%); cohort 2: *n* = 664 (81.6%); b = −0.03 (*p* < 0.0001) for the first cohort, and b = −0.10 (*p* < 0.0001) for the second cohort), but another portion reported an increase (cohort 1: *n* = 83 (17.4%); cohort 2: *n* = 150 (18.4%); b = 0.32 (*p* < 0.0001) in the first cohort, and b = 0.47 (*p* < 0.0001) in the second cohort). This difference seems to be due to the presence of high levels of psychological problems for the group that reported low positive affects and for the group that reported high negative affects.

In the second study [87], conducted post-pandemic, the authors recorded the effects of the app on adolescents’ affectivity and cognitive well-being, depressive symptoms, anxiety, and the impact of COVID-19 through online questionnaires. After playing the Grow It! app, affective well-being significantly increased in both cohort 1 (t = −6.806, *p* < 0.001, and d = 0.32) and cohort 2 (t = −6.77, *p* < 0.001, and d = 0.23). Similarly, cognitive well-being improved significantly in cohort 1 (t = −6.12, *p* < 0.001, and d = 0.27) and cohort 2 (t = −5.93, *p* < 0.001, and d = 0.20) [14]. Depressive symptoms showed a significant decrease in both cohorts (cohort 1: t = −2.91, *p* = 0.004, and d = 0.08; cohort 2: t = −7.34, *p* < 0.001, and d = 0.17). Anxiety significantly decreased in cohort 2 (t (732) = −4.69, *p* < 0.001, and d = 0.14) but not in cohort 1.

## 4. Discussion

The current systematic review aims to identify and analyze the digital tools, in particular serious games, that have been developed to promote well-being and empowerment in adolescents and young adults, with a specific focus on gender perspectives. Following the rigorous Preferred Reporting Items for Systematic Reviews (PRISMA) guidelines, the authors outline a methodical approach to their investigation [77,78]. This structured approach not only helps refine the scope of the review but also ensures clarity and precision in the search strategy.

The article selection process resulted in the inclusion of 15 articles. The transparent presentation of the screening process in Figure 1 increases the credibility of the study by providing readers with a visual representation of the record selection steps.

The systematic review explores various aspects of well-being, mental health, empowerment, and physical and sexual health in adolescents and young adults. Its aim is to provide insights into the multifaceted nature of development within this population. The World Health Organization’s definition of well-being, which encompasses physical, mental, and social dimensions, serves as the foundation for understanding the intricate interplay of factors that contribute to a balanced and positive life experience for adolescents and young adults.

Numerous efforts have been made to explore the use of digital tools to enhance the effectiveness of psycho-educational interventions aimed at promoting protective factors for perceived well-being, considering different aspects.

Among the aspects considered in our systematic review on psychological well-being among adolescents and young adults, we considered the cognitive aspect to be relevant. Already Bickman et al. [80], through the elaboration of the Screenshots program following the introduction of full-time online education and technology-mediated communication, noted how increasing knowledge about key concepts of digital citizenship resulted in changing beliefs and behaviors to preserve prosocial and safe online interactions. Furthermore, in line with our systematic review, they highlighted that the Screenshots program showed gender differences in the way adolescents experience and perpetrate online aggression.

Promising results also emerged regarding the use of digital tools in the promotion of emotional and behavioral adaptations that may foster the affective components of well-being, which are related to personal growth and self-determination. We analyzed, in this sense, the intervention of Cejudo et al. [82] on the video game “Aislados”. Specifically, school performance, reduction in personal risk behavior, and increase in adaptive coping strategies were linked to increased health-related quality of life and positive affectivity, indicating the potential of innovative approaches, such as gamified interventions, in addressing emotional well-being in adolescents.

Moreover, in applying the criteria for our systematic review, we extracted the work by Cheng et al. [83] on the use of the MindMax app. MindMax addressed barriers to mental health help seeking, with a particular focus on men. Factors that appeared conducive to psycho-educational intervention were closely related to the user-centered design of the app, such as the varied presentation of information, balanced use of formal and informal language, and active engagement. As suggested by our review, MindMax with its innovative approach to mental health and well-being education, focusing on men aged 16–35 years, increased the effectiveness of the psycho-educational intervention. The pilot study, conducted in two schools in Santiago, Chile, and two schools in Medellín, Colombia, by Martinez et al. [89] showed a high level of acceptance of depression in adolescence. The application of an Internet-based program, “Cuida tu Ánimo” (CTA) (Take care of your soul), for the prevention and early intervention of depression in adolescence improved intervention outcomes and adherence levels. The different impact of the “Cuida tu Ánimo” program in Chile and Colombia, as observed, underlined contextual influences on mental health interventions. Customization and context-specific strategies are imperative to optimize intervention outcomes and adherence.

In line with our considerations regarding the need to recognize gender differences in the effectiveness of interventions that use digital tools and serious games to promote psychological well-being and empowerment, Wiljer et al. [94], through the evaluation of the Thought Spot application, provided indications of gender-specific patterns in help-seeking behavior among post-secondary students. In particular, mHealth interventions appear to increase help-seeking intentions similarly to information brochures. The literature suggests a qualitative difference between genders in their reliance on formal and informal help-seeking resources. Women tend to seek help from formal resources more often than men, while the latter are more likely to seek help from informal sources. Recognizing these nuances is crucial for designing interventions that are in line with the different needs and preferences of male and female adolescents.

Jiang et al. [97] suggested that prevention and intervention programs should address adolescents’ perceived attachment with parents. These training programs should improve communication between parents and their adolescent children. Indeed, the promotion of empowerment, as a pathway to well-being, has been the subject of interest of numerous scholars. The authors [97] identified goal setting and various pathway strategies within learning contexts, such as regular classroom activities as interpersonal and environmental variables, as successful strategies in developing hopeful thinking and enhancing subjective well-being.

Oberle et al. [98], studying life satisfaction in relation to contextual ecological variables such as social and emotional well-being and prosperity in early adolescence, argued that positive and supportive relationships with non-community adults, along with a strong sense of belonging at school, may be critical in determining satisfaction or happiness. In some cases, as in the study by Ravaccia et al. [93], gender differences in the use of digital tools were not considered relevant. Instead, anonymity in the use of a digital peer support tool named MeeToo highlighted several positive effects, including the ease of talking about difficult issues, being part of a supportive community, finding new ways to help, feeling better, and feeling less alone. On the other hand, the study by Denton-Calabrese et al. [86] highlighted gender differences in the use of digital tools, media production, and coding.

The go_girl program sought to counter influences that had a negative impact on the academic self-esteem of marginalized young women. Academic self-concept, technological empowerment, and social empowerment are interconnected in such a way that learning through digital technologies, in a personalized and supportive environment, can increase one’s sense of mastery and aspirational ability. D’Aprile et al.’s [84] investigation of the serious health game “Tako Dojo” revealed connections between gaming experience, empowerment, and therapeutic adherence among young people with diabetes. Gender-specific nuances underline the importance of tailoring interventions to the unique needs of male and female participants. Cognitive abilities, skills, attitudes, and usage patterns of serious games between genders [99] could be considered as relevant factors and behaviors that can enhance engagement in gaming and, consequently, game-based learning activities for both males and females. Therefore, we believe that gender differences must be taken in consideration when designing and developing serious games for health.

Our review considered the dimensions of physical and sexual health as dimensions of well-being. As highlighted by Haruna et al. [88], of particular effectiveness is the use of gamification to effectively improve sexual health knowledge by overcoming taboos for adolescents vulnerable to high-risk sexual behavior and activating HIV/AIDS prevention interventions. Providing knowledge on topics traditionally considered taboos is an avenue to be pursued with determination. The integration of technology in health education shows the potential of digital tools in mitigating chronic diseases and promoting healthy lifestyles. An effective example of this is the ePsiconut program [81], a psycho-nutritional program fully supported by eHealth tools. In particular, the use of this tool enabled the promotion of the consumption of food of high nutritional quality. Taking care of nutritional health, in line with the observations of our review, allowed for an increase in subjective well-being and a reduction or better management of anxiety-depressive symptoms. Furthermore, the qualitative study by Morgan et al. [91] sheds light on the therapeutic role of avatars in the well-being of transgender and gender-diverse youth. Acting as a “safe place” for self-expression and exploration of gender identity, avatars emphasize the importance of inclusive and affirming digital spaces for adolescents facing complex identity paths.

This review also explored studies that have used well-being promotion tools that fall within the frame of Positive Psychology. In line with the literature [96], digital tools can be very useful for developing individuals’ internal resources, such as their self-efficacy, self-determination, and sense of personal satisfaction, also through the reduction of negative thoughts. This would seem to be true especially for women [92], although it is also evident that gender differences in this perspective need further investigation.

Finally, our review also addressed the question of the effectiveness of using digital tools integrated with CBT in adolescents after the COVID-19 pandemic [87]. Longitudinal studies by Mens et al. [90] and Dietvorst et al. [87] demonstrated the impact of the “Grow It!” application integrated with CBT on affective and cognitive well-being, depressive symptoms, anxiety, and COVID-19 pandemic repercussions. It appears that the improvement brought about by the “Grow It!” application integrated with CBT on daily well-being is significant. However, for individuals at higher risk of developing mental health problems, the integration of the “Grow It!” app with CBT showed a significant decrease in daily well-being. Therefore, these considerations reinforce a key idea of our review: distinguishing between the use of digital tools in psycho-educational versus clinical settings. The gamified Cognitive Behavioral Therapy approach is promising for the prevention of mental health problems in adolescents; less stable is its effectiveness for clinical contexts. Therefore, the integration of the “Grow It!” application with CBT reinforced the link between adaptive coping and positive affects, suggesting that coping helps adolescents to be resilient in times of stress.

Combining the findings of these different studies highlights the importance of holistic, context-aware, and inclusive approaches to promote adolescent well-being. It emphasizes the need to continue to innovate, to tailor interventions to specific demographic and cultural contexts, and to harness the potential of technology and gamification to address the multiple challenges adolescents face on their path to well-being and positive mental health.

Considering the powerful role of serious games and digital tools and their widespread use among adolescents and young adults, the review highlights that while there are many applications of these tools for well-being, only a few currently address and intervene on important developmental issues such as the gender gap, inequality, and identity.

However, these findings need to be considered in light of different ways in which the construct of well-being has been operationalized, along with the various methodologies employed to measure it. This makes it difficult to compare the effectiveness of the digital tools analyzed, thus introducing further complexity to the phenomenon. It is surprising that subjective well-being measures are used without adding context-level variables.

The literature is fairly consistent in emphasizing the importance of including interpersonal and environmental variables in the study of subjective well-being [97,98,100,101]. This implies that there is a critical need for more focused research in the future, designed explicitly to investigate and address the nuanced ways in which serious games and digital tools may impact the well-being and empowerment of adolescents and young adults, with due consideration of gender dynamics in the most inclusive possible way. While some studies have acknowledged gender differences or focused on predominantly female cohorts, a more deliberate and comprehensive approach to understanding the distinct experiences and needs of genders within digital well-being interventions is essential.

## 5. Conclusions

In conclusion, while the systematic review provides a solid basis for understanding the landscape of digital tools for promoting well-being, it also highlights the existing gap in the current literature regarding gender-specific considerations. Addressing this gap in future research will not only contribute to a more comprehensive understanding of the impact of serious games on the well-being of adolescents and young adults but will also pave the way for more targeted and inclusive interventions in line with different gender experiences and needs. Furthermore, review and opinion studies were excluded from the review, as the aim was to systematize findings from evidence-based studies. The combination of findings from these different studies highlights the importance of holistic, context-aware, and inclusive approaches to promote adolescent well-being.

Researchers should also aim to develop and implement interventions that explicitly address the unique well-being challenges faced by individuals across the gender spectrum. By doing so, the field can move beyond recognizing general gender differences and work toward creating tailored inclusive, equitable, and effective interventions for various gender identities. The systematic review underscores the critical importance of adopting holistic, context-aware, and inclusive strategies for promoting adolescent well-being, while also emphasizing the need for continued innovation and culturally tailored interventions. Despite the valuable insights provided, a significant gap remains in the literature regarding gender-specific considerations, indicating a crucial area for future research to address in order to enhance the effectiveness and inclusivity of digital tools for well-being promotion.

Future research efforts should incorporate a more refined exploration of gender-specific outcomes, delving into the differential effects of serious games on the psychological well-being and empowerment of males and females. This not only requires a more balanced representation of gender in the study samples but also requires a more nuanced analysis of how these interventions may intersect with and influence different gender identities.

### Limitations

This review acknowledges several limitations. One limitation cuts across all systematic reviews, namely that the research findings are limited by the search terms and refinements used (e.g., the journals included and the time period of publications). Although the systematic review may not accurately reflect all the existing literature relevant to this study, it does provide insight into current research findings and the impact of the inclusion of games in intervention programs. Indeed, the study’s results led to the selection of only 15 articles that satisfied all the required criteria. This implies that the study offers a limited view of what is present in the literature on the investigated topic.

Another limitation is that despite PRISMA’s quality criteria and the authors’ adherence to guidelines to ensure a certain methodological rigor, the authors cannot fully control publication bias and, thus, cannot guarantee full access to the data within this systematic review. Thus, our review has limitations regarding the quality of the included studies.

Some of the studies considered may have been characterized by a risk of bias towards the use of digital tools, especially in the clinical setting. Other studies considered may have limitations due to the unclear handling of confounding factors or missing data. Furthermore, as not all studies described the reliability and validity of the tools used to measure knowledge, we cannot assume that outcome measurements are highly reliable. Despite these limitations, this review provides valuable insights for educators and researchers to use new and innovative approaches in promoting well-being by synthesizing current evidence for the construction of content for serious games and the technological aspects of serious games applications.

Another limitation concerns the inclusion of only one study that investigated the well-being of the gender diverse and gender non-conforming population. Considering the gender perspective in which the review was conducted, an important issue that emerges from the results concerns the fact that among the included studies, few explicitly considered the promotion of well-being from a gender perspective, even if they considered differences between men and women or recruited a predominantly female sample.

Finally, these results must be considered with the understanding that limitations arise from the various ways in which the construct of well-being was operationalized across the different articles. The diverse methodologies employed to measure well-being introduce additional complexity, making it challenging to compare the effectiveness of the analyzed digital tools.

## Figures and Tables

**Figure 1 behavsci-14-01052-f001:**
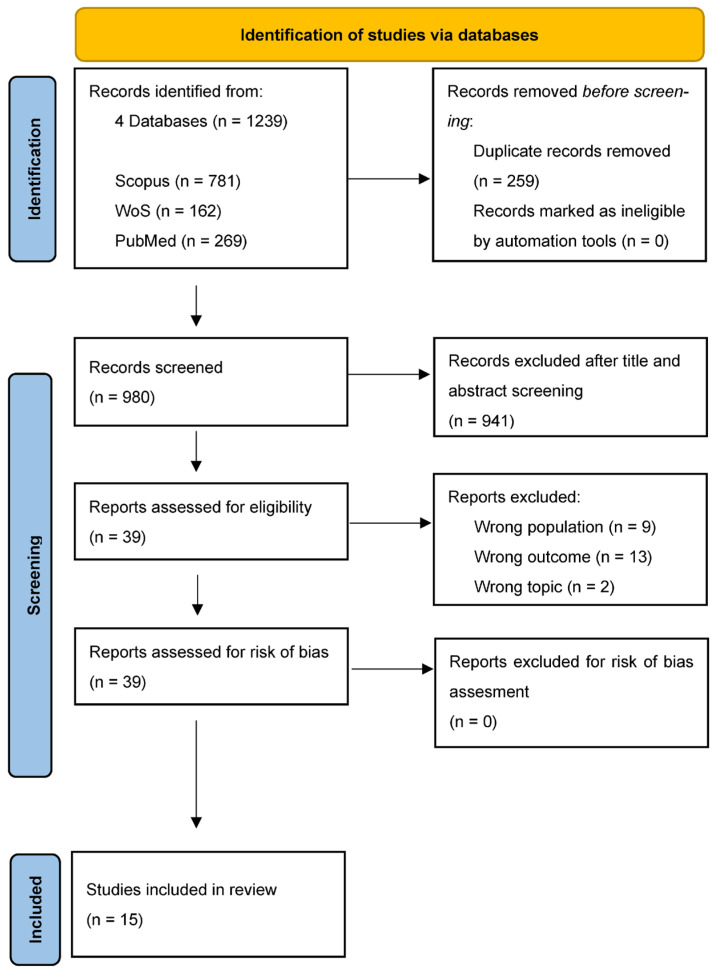
PRISMA flowchart showing the selection process conducted.

**Table 1 behavsci-14-01052-t001:** Overview of studies included in the systematic review.

Authors	Study Design	Country and Samples	Digital Tool	Main Dimension of Well-Being Considered	Main Results
Bickham et al. [80]	Quasi-experimental study	USA (New England) Experimental group: *n* = 92 students; control group: *n* = 71 students (66% females; (age: M = 12; SD = 0.4)	Screenshots: an in-school curriculum that imparts knowledge and teaches social skills, improving young people’s well-being and health	Psychological well-being (cognitive component)	The results revealed a rise (F = 9.97; *p* = 0.003) in understanding of essential curriculum concepts. The participants reported a reduction (F = 9.00; *p* = 0.004) in their endorsement of discourteous online conduct and an elevation (F = 4.39; *p* = 0.04) in their support for online safety and privacy practices. Significant gender differences were observed: boys demonstrated a decrease (F = 5.77; *p* = 0.006) in the likelihood of employing verbal aggression and an increase (F = 3.65; *p* = 0.04) in the adoption of a more prosocial approach to resolving conflicts.
Cantisano et al. [81]	Pre–post study	Dominican Republic *n* = 16 female university students (age: M = 20.69; SD = 1.74)	ePSICONUT: an online program created to promote healthy lifestyle habits in university students as well as psychological status	Physical health	The results showed a significant enhancement in the participants’ overall dietary quality post-ePSICONUT program (t = 0.92; *p* < 0.05), characterized by an increased consumption of vegetables (*p* = 0.005) and a decrease in the intake of sugary drinks (*p* = 0.042). Additionally, the program led to a statistically significant improvement in healthy lifestyles (t = 3.02, *p* = 0.009). Subjective well-being significantly increased (*p* < 0.05), and anxiety levels decreased (*p* < 0.05). The participants perceived a significant impact of the program on their subjective well-being (M = 4.38; SD = 0.72; range: [1,2,3,4,5]).
Cejudo et al. [82]	Quasi-experimental study	Spain Adolescents *n* = 187 Female = 53%	Aislados (Isolated): a video game for the improvement of well-being and trait emotional intelligence of adolescents	Psychological well-being (cognitive and affective components)	The results showed a significant increase (F = 1.99, *p* = 0.038, and d = 0.22) in health-related quality of life related to the cognitive component of well-being and a significant increase (F = 2.932, *p* = 0.041, and d = 0.38) in positive affects related to the affective component of well-being.
Cheng et al. [83]	Qualitative study	Australia *n* = 15 young adults (67% male; age: [16; 35])	MindMax: an app that incorporates gamification and mini-games	Mental health	The results indicate a favorable response from users regarding the incorporation of mental health, well-being, sports, and video games into the MindMax platform. The users preferred diverse information delivery methods, including video subtitles and transcripts, as they tended to skip videos. The users also showed sensitivity to the tone used in health and well-being apps, finding a formal tone intimidating while perceiving an informal tone as less imposing but potentially diminishing the seriousness of the content. Active engagement and the application of practical skills were also favored by users.
D’Aprile et al. [84]	Cross-sectional study	Italy *n* = 124 young adults (60% females; age: M = 14.12, SD = 2.14)	Tako Dojo: a digital serious game developed to improve the quality of life of young people with chronic diabetes	Empowerment	The results revealed significant correlations between gender-related behaviors and adherence to diabetes therapy (r = 0.19; *p* < 0.05), between game experience and empowerment in managing diabetes (r = 0.28; *p* < 0.01), and between empowerment in diabetes management and behaviors related to therapy adherence (r = 0.52; *p* < 0.01). Engaging in the serious game significantly increased empowerment in diabetes management among participants (β = 0.88; *p* < 0.01) and empowerment had a positive impact on behaviors related to therapeutic adherence for diabetes (β = 0.46; *p* < 0.01). In terms of gender differences, males experienced higher levels (β = 1.88; *p* < 0.01) of empowerment through the serious game. This heightened empowerment in males had both direct (β = 1.15; *p* < 0.01) and indirect (β = 0.55; *p* < 0.01) positive effects on therapeutic adherence. For female participants, a positive relationship was identified between empowerment and behaviors related to therapy adherence (β = 0.46; *p* < 0.01).
Daugherty et al. [85]	Quasi-experimental study	Indiana *n* = 112 young adults (70.5% females; age: range [18,19,20,21,22,23,24,25])	RealLife EXP: a free smartphone app designed to foster hope and promote hedonic (HWB) and eudaimonic well-being (EWB)	Well-being from a Positive Psychology perspective	The results reported significant changes in hope across the intervention and control conditions (F = 4.24; *p* = 0.05). There was no difference in hedonic (HWB) and eudaimonic (EWB) well-being between intervention and control groups. No differences were found regarding gender.
Denton-Calabrese et al. [86]	Qualitative study	UK 9 marginalized NEET young women (age: [16,17,18,19,20,21])	Go_girl: code + create: a community outreach program that aims to empower young women who are NEET	Empowerment	The participants expressed that their self-concept in terms of education and career aspirations was impacted by the program. One participant set higher goals, three gained more clarity about their goals, and two acquired a broader perspective on what they can achieve in life. Among the participants, five noted shifts in their self-concept regarding computing, seeing themselves as creators rather than just consumers of technology. Many participants expressed satisfaction with the hands-on learning approach, and a notable outcome concerning self-concept was the recognition of their ability to use technology for creative purposes.
Dietvorst et al. [87]	Longitudinal cohort study	Deutschland Cohort 1: *n* = 1282 adolescents (68% girls; age: M = 16.67 years, SD = 3.07); cohort 2: *n* = 1871 adolescents (81% girls; age: M = 18.66 years, SD = 3.7)	Grow it! App: an mHealth digital intervention aimed at preventing mental health problems	COVID-19 pandemic-related well-being	The results indicate that following engagement with the Grow It! app, there was a statistically significant increase in affective well-being in both cohort 1 (t = −6.806, *p* < 0.001, and d = 0.32) and cohort 2 (t = −6.77, *p* < 0.001, and d = 0.23). Cognitive well-being also demonstrated a statistically significant improvement from baseline to follow-up in both cohort 1 (t = −6.12, *p* < 0.001, and d = 0.27) and cohort 2 (t = −5.93, *p* < 0.001, and d = 0.20). Furthermore, depressive symptoms showed a significant decrease from baseline to follow-up in both cohorts (cohort 1: t = −2.91, *p* = 0.004, and d = 0.08; cohort 2: t = −7.34, *p* < 0.001, and d = 0.17), while anxiety significantly decreased just in cohort 2 (t = −4.69, *p* < 0.001, and d = 0.14).
Haruna et al. [88]	Mixed-methods study	Tanzania *n* = 120 lower secondary school students (48% females; age: range [11,12,13,14,15])	Game-Based Learning (GBL), gamification, and traditional teaching	Sexual health	The results reported that the scores in the post-test for Game-Based Learning (GBL) (M = 79.94; SD = 11.169) and gamification (M = 79.23; SD = 9.186) were significantly higher (F = 54.75; *p* = 0.001) compared to the control group (M = 51.93; SD = 18.705). Significant differences (*p* ≤ 0.05) were observed across the motivation, attitude, knowledge, and engagement constructs, Statistically significant differences were found for the constructions of motivation (*p* = 0.001), attitude (*p* = 0.001), knowledge (*p* = 0.001), and engagement (*p* = 0.001), indicating that the GBL and gamification experimental settings had a notably positive impact on students’ understanding and acquisition of sexual health knowledge, surpassing those in the traditional setting.
Martínez et al. [89]	Mixed-methods study	Colombia and Chile *n* = 199 adolescents (73% from Chile; 53.3% females; age: M = 14.8, SD = 1.0)	Cuida tu Ánimo (Take care of your mood): an internet-based program for the prevention and early intervention of adolescent depression	Mental health	The results indicated a decrease in depressive and anxious symptoms among adolescents in Chile (*p* < 0.001). The sex of the participants was a predictor of website use, with women being more likely to use it extensively (OR = 2.9). The participants’ level of satisfaction with using the app was high (82.5%). The participants perceived the program as a resource for promoting emotional well-being and a space for uninhibited self-expression, also voicing a need for increased engagement with the research team.
Mens et al. [90]	Longitudinal cohort study	Deutschland Cohort 1: *n* = 476 adolescents and young adults (76% females; age: M = 16.24, range [12,13,14,15,16,17,18,19,20,21,22,23,24,25]); cohort 2: *n* = 814 adolescents and young adults (83% females; age: M = 18.45, range [12,13,14,15,16,17,18,19,20,21,22,23,24,25])	Grow it! App: an mHealth digital intervention, combining the Experience Sampling Method (ESM) with gamified Cognitive Behavioral Therapy (CBT), aimed at preventing mental health problems	COVID-19 pandemic-related well-being	The results showed an increase in positive affects over time for some participants (cohort 1: *n* = 308 (64.7%); cohort 2: *n* = 586 (72.0%); b = 0.06 (*p* < 0.0001) for the first cohort, and b = 0.10 (*p* < 0.001) for the second cohort), while others experienced a decrease in positive affects (cohort 1: *n* = 168 (35.3%); cohort 2: *n* = 228 (28.0%); b = −0.27 (*p* < 0.0001) for the first cohort, and b = −0.52 (*p* < 0.0001) for the second cohort). Similar trends were observed for negative affects, with some participants reporting a decline (cohort 1: *n* = 393 (82.6%); cohort 2: *n* = 664 (81.6%); b = −0.03 (*p* < 0.0001) for the first cohort, and b = −0.10 (*p* < 0.0001) for the second cohort), while others noted an increase (cohort 1: *n* = 83 (17.4%); cohort 2: *n* = 150 (18.4%); b = 0.32 (*p* < 0.0001) in the first cohort, and b = 0.47 (*p* < 0.0001) in the second cohort).
Morgan et al. [91]	Qualitative study	Australia *n* = 17 TGD young people (age: [11,12,13,14,15,16,17,18,19,20,21,22])	Avatars in games: used to explore and represent their gender identity	Psychological well-being (affective component)	According to participants, the act of creating or choosing an avatar that mirrors their perceived gender provides a “safe space” to initiate the exploration and comprehension of their gender identity prior to doing so in the physical realm. The avatar played a significant role not just in comprehension but also in “fine-tuning” the participants’ understanding of their gender identity. The participants expressed positive feelings when engaging with an avatar crafted to represent their perceived gender, viewing it as the sole means of expressing their gender before openly acknowledging it. They attributed a potentially therapeutic function to this practice during emotionally challenging periods.
Parks et al. [92]	Pre–post study	Participants from many countries around the world; *n* = 821 users with an age higher than 18 (82% females)	Happify: a digital well-being program addressing issues, such as overcoming negative thoughts, through games and activities grounded in positive psychology, Cognitive Behavioral Therapy, and mindfulness principles	Well-being from a Positive Psychology perspective	The results suggest that individuals, regardless of whether they have a chronic condition or not, experienced similar enhancements in well-being through their engagement with Happify and by increasing their activity levels on the platform. Both groups displayed consistent improvements in well-being over time. Furthermore, the analysis revealed significant effects for both the duration of engagement since baseline (b = 0.071; *p* < 0.01) and the quantity of activities completed (b = 0.03; *p* < 0.01). Those who had been active participants on Happify for a longer duration and had completed more activities tended to report higher levels of subjective well-being. Additionally, a significant interaction emerged between the number of activities completed and the duration since baseline (b = 0.0002; *p* < 0.01).
Ravaccia et al. [93]	Mixed-methods study	UK *n* = 876 young people (76% female; age: [11,12,13,14,15,16,17,18,19])	Tellmi (ex MeeToo): an anonymous, fully moderated peer support tool for young people’s empowerment and well-being	Empowerment	The results indicate a significant improvement (t = 2.15; *p* = 0.04) in the overall well-being of female participants after utilizing the tool. The participants highlighted that the anonymity provided by the tool allowed them to freely express themselves, connect with others who shared similar experiences, and feel empowered in offering support to fellow participants.
Wiljer et al. [94]	Randomized controlled trials (RCTs)	Canada (Toronto area) Treatment group: *n* = 241 (79% females; age: M = 22.9, SD = 3.4); control group: *n* = 240 (78% females; age: M = 23.2, SD = 3.1)	Thought Spot: mobile and web app created to promote help-seeking intentions and well-being of post-secondary students	Mental health	The results reported a noteworthy time effect (F = 23.52; *p* < 0.001; f = 0.21). Gender had a significant impact on help-seeking behavior from formal and/or informal sources: women exhibited higher scores in formal help seeking (OR = 1.86; *p* = 0.001) and lower scores in informal help seeking (OR = 0.80; *p* < 0.001); moreover, women displayed lower scores in the inclination to seek professional psychological help (OR = 0.80; 95%, *p* = 0.003).

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
