# Peer review of "Promoting Well-Being from a Gender Perspective: A Systematic Review of Interventions Using Digital Tools and Serious Games"

_behavsci, 2024, doi:10.3390/bs14111052_

Round 1

Reviewer 1 Report

Comments and Suggestions for Authors

The article provides a very detailed review of the tools that can be used to promote the well-being of adolescents from a gender perspective. It is well-written and well-cited. I recommend only a few areas of potential improvement below.

As this paper includes extensive details it would be helpful to outline the categories/aspects around which the analysis was based: Cognitive and affective components of well-being, mental health, empowerment, physical and sexual health, positive psychology in the ‘Aim’ section to help readers organize their expectations for the rest of the paper. Ideally these categories/aspects would be noted again in the Discussion with a brief (one – two sentence) recommendation regarding what appears to work well and what doesn’t. These are the actionable insights that the reader will be most likely looking for and will greatly improve the usefulness and impact of this article.

For the section beginning on line 323 it would be interesting to include whether all studies included options for participants to identify as genders other than male and female.

For the section between line 430 and 435 please include evidence of the ‘influence’ and whether it was positive or negative, as well as evidence regarding the reported changes in self-concept.

Comments on the Quality of English Language

There are a few minor grammar issues, but that is all.

Author Response

Reviewer 1

“As this paper includes extensive details it would be helpful to outline the categories/aspects around which the analysis was based: Cognitive and affective components of well-being, mental health, empowerment, physical and sexual health, positive psychology in the 'Aim ' section to help readers organize their expectations for the rest of the paper. Ideally these categories/aspects would be noted again in the Discussion with a brief (one – two sentence) recommendation regarding what appears to work well and what doesn't. These are the actionable insights that the reader will be most likely looking for and will greatly improve the usefulness and impact of this article.”

Response: Thank you for your note. As suggested by the reviewer, we listed the wellness aspects in the "aim" section (lines 203-208). Additionally, we revised the Discussion to address the results related to these categories in a more linear way.

“For the section beginning online 323 it would be interesting to include whether all studies included options for participants to identify as genders other than male and female.”

Response: Thank you for the suggestion. We added a sentence that goes into more detail on this aspect (lines 333-336).

“For the section between line 430 and 435 please include evidence of the 'influence' and whether it was positive or negative, as well as evidence regarding the reported changes in self-concept.”

Response: Thank you for this comment. We did not include numerical evidence, because the study is qualitative and presents only results reported by participants. However, to address the misunderstanding, we have revised the "too causal" language, now using "connection" instead of "influence". We have also clarified that this connection is positive (lines 438-441).

“There are a few minor grammar issues, but that is all.”

Response: We agree with the reviewer. We have further revised the English throughout the paper for clarity and readability.

Reviewer 2 Report

Comments and Suggestions for Authors

The selection of articles looks like proof that the topic is not covered and there is not enough research to form an idea - and not like a scientific methodology. A study of 15 articles is not a review paper; any ordinary empirical study uses many more articles to present the topic.

Furthermore, the task of a review article is not a simple retelling of articles, but their deep analysis as a whole, which would allow reaching a new level of understanding. In this study, first, a description of the articles is given in a table (pp. 8-13) - then it begins to be repeated in the text (pp. 14-18). What is the point of this?

In general, the topic being studied is interesting, but it is almost impossible to make research from a retelling of 15 works. But in this case, authors did not even try - they simply made a review of the articles, this is not the level of publication in a serious journal. It would be advantageous to base own digital technology project, aimed at solving the problem based on existing experience. In principle, the authors can try to do theoretical work, but it is more difficult, and you need to come to new knowledge.

Author Response

Reviewer 2

“The selection of articles looks like proof that the topic is not covered and there is not enough research to form an idea - and not like a scientific methodology. A study of 15 articles is not a review paper; any ordinary empirical study uses many more articles to present the topic.”

Response: Thank you for the comment. We identified the study as a systematic review because we strictly followed the procedure provided for this methodology. We selected only 15 articles, precisely to adhere closely to these guidelines. Additionally, we have addressed this aspect in the limitations of the study (770-772).

“Furthermore, the task of a review article is not a simple retelling of articles, but their deep analysis as a whole, which would allow reaching a new level of understanding. In this study, first, a description of the articles is given in a table (pp. 8-13) - then it begins to be repeated in the text (pp. 14-18). What is the point of this?”

Response: Thank you for the advice. We found your comments extremely helpful and revised accordingly. In the text we have summarized the results that emerged from the selected articles, which are described in more detail in a table, as is customary in a systematic review. Regarding the critical analysis of these findings, we have further revised and expanded in the “discussions” (lines 583-735).

“In general, the topic being studied is interesting, but it is almost impossible to make research from a retelling of 15 works. But in this case, authors did not even try - they simply made a review of the articles, this is not the level of publication in a serious journal. It would be advantageous to base own digital technology project, aimed at solving the problem based on existing experience. In principle, the authors can try to do theoretical work, but it is more difficult, and you need to come to new knowledge.”

Response:Thank you for your claim. This work aims to provide useful information about the digital tools developed and used to promote psychological well-being from gender perspective. While it certainly has a theoretical purpose, it also offers practical insights for readers interested in the topic who wish to explore the literature to create or improve a digital tool. However, this is not our primary intention. As mentioned earlier, we have highlighted the selection of only 15 articles as a limitation of the study (770-772).

Reviewer 3 Report

Comments and Suggestions for Authors

The paper addresses gender inequalities and more generally, induced well-being issues and how computer-based tools can help mitigate or correct biases and change behaviors. These disparities come from gender stereotypes which cultural roots arise since childhood from school, parents, media or other sources. This research work is based on a systematic review of state-of-the-art studies using digital tools, including serious games, to promote well-being of young people, from a gender perspective.

The method followed for the systematic review was driven by the PRISMA guidelines. The final material was made of 15 research articles written in English published between 2018 to 2022. They have been selected from a set of 1239 collected articles from four databases: Scopus, Web of Science, PubMed and PsycInfo, using a multi-step filtering according to different criteria. 

The method is well described, the method is relevant, the choice refinement is clearly explained and a flow-chart shows the selection process and gives a good synthesis. The low number of final papers and the selection criteria may produce some bias, that is discussed later in the paper. As example, topics such as computer science or science of education have not been considered as fields of study, but only as technical keywords. Nevertheless, papers using serious or video games or other computer support have been finally selected.

For each of the 15 papers, a lot of raw data (such as authors, study design, country and samples, digital tool, main dimension of well-being considered and main results) have been collected. Even if 15 papers can be considered as low number, they cover several topics, and bring out a multitude of platforms, applications, and programs aimed at promoting well-being, addressing lots of application fields including emotional, cognitive, or mental, physical and sexual health dimensions. 

Some stats are given after the raw results. Papers are then analyzed more deeply according to more focused topics (Cognitive and affective components of well-being, Mental health, Empowerment, Physical and sexual health, Positive Psychology), but because of these diversities of application fields, digital tools and set-ups, it is difficult to generalize to have result discussions and to make conclusions. 

The discussion points out more effective solutions among propositions exposed upper from selected paper and discus pro and cons and some particular aspects relative to gender or well-being. The powerful role of serious games and digital tools is underlined in the study context, fostered by the interest of younger people for using digital games/tools. Gender differences noted in the studied papers are also commented.

This systematic review aims to the expanding knowledge base on technology-based interventions for social change. It endeavours to empower educators and advance the creation of innovative, evidence-based digital tools that can enhance positive mental health, promote gender equality education, and foster the well-being of young people.

Some limitations are pointed out: few terms were used for the paper quest, the refinements process could be improved, the papers set is not exhaustive, selected papers can have their own limitations, ... This can produce bias in the discussion and conclusion but their main lines are coherent and remain relevant. As mentioned by the authors, future research should include a more refined exploration of gender-specific outcomes, looking into the differential effects of serious games on the psychological well-being and empowerment of males and females. This should lead to have a more balanced representation of gender in the study samples, and also should require a better analysis of how this can affect and influence different gender identities.

As a conclusion, the main outcome of this paper is to bring guidelines other researchers to ask themselves good questions to design studies for their own applications in these targeted fields using digital tools and this can help them to focus directly to what can interest them more directly.

Author Response

Reviewer 3

The paper addresses gender inequalities and more generally, induced well-being issues and how computer-based tools can help mitigate or correct biases and change behaviors. These disparities come from gender stereotypes which cultural roots arise since childhood from school, parents, media or other sources. This research work is based on a systematic review of state-of-the-art studies using digital tools, including serious games, to promote well-being of young people, from a gender perspective.

The method followed for the systematic review was driven by the PRISMA guidelines. The final material was made of 15 research articles written in English published between 2018 to 2022. They have been selected from a set of 1239 collected articles from four databases: Scopus, Web of Science, PubMed and PsycInfo, using a multi-step filtering according to different criteria. 

The method is well described, the method is relevant, the choice refinement is clearly explained and a flow-chart shows the selection process and gives a good synthesis. The low number of final papers and the selection criteria may produce some bias, that is discussed later in the paper. As example, topics such as computer science or science of education have not been considered as fields of study, but only as technical keywords. Nevertheless, papers using serious or video games or other computer support have been finally selected.

For each of the 15 papers, a lot of raw data (such as authors, study design, country and samples, digital tool, main dimension of well-being considered and main results) have been collected. Even if 15 papers can be considered as low number, they cover several topics, and bring out a multitude of platforms, applications, and programs aimed at promoting well-being, addressing lots of application fields including emotional, cognitive, or mental, physical and sexual health dimensions. 

Some stats are given after the raw results. Papers are then analyzed more deeply according to more focused topics (Cognitive and affective components of well-being, Mental health, Empowerment, Physical and sexual health, Positive Psychology), but because of these diversities of application fields, digital tools and set-ups, it is difficult to generalize to have result discussions and to make conclusions. 

The discussion points out more effective solutions among propositions exposed upper from selected paper and discus pro and cons and some particular aspects relative to gender or well-being. The powerful role of serious games and digital tools is underlined in the study context, fostered by the interest of younger people for using digital games/tools. Gender differences noted in the studied papers are also commented.

This systematic review aims to the expanding knowledge base on technology-based interventions for social change. It endeavours to empower educators and advance the creation of innovative, evidence-based digital tools that can enhance positive mental health, promote gender equality education, and foster the well-being of young people.

Some limitations are pointed out: few terms were used for the paper quest, the refinements process could be improved, the papers set is not exhaustive, selected papers can have their own limitations, ... This can produce bias in the discussion and conclusion but their main lines are coherent and remain relevant. As mentioned by the authors, future research should include a more refined exploration of gender-specific outcomes, looking into the differential effects of serious games on the psychological well-being and empowerment of males and females. This should lead to have a more balanced representation of gender in the study samples, and also should require a better analysis of how this can affect and influence different gender identities.

As a conclusion, the main outcome of this paper is to bring guidelines other researchers to ask themselves good questions to design studies for their own applications in these targeted fields using digital tools and this can help them to focus directly to what can interest them more directly.

Response: We thank the reviewer for examining our manuscript so carefully and hope that the additions we made now have further improved the paper.